# Optical imaging of flavor order in flat band graphene

Tian Xie [1], Tobias M. Wolf[2], Siyuan Xu[1], Zhiyuan Cui[1], Richen Xiong[1], Yunbo Ou [3], Patrick Hays[3], Ludwig F. Holleis [1], Yi Guo[1], Owen I. Sheekey[1], Caitlin Patterson[1], Trevor Arp[1], Kenji Watanabe [4], Takashi Taniguchi [5], Seth Ariel Tongay [3], Andrea F. Young[1], Allan H. MacDonald [2] ✉ & Chenhao Jin [1] ✉

Spin- and valley flavor polarization plays a central role in the many-body physics of flat band graphene, with Fermi surface reconstruction — often accompanied by quantized anomalous Hall and superconducting state — observed in a variety of experimental systems. Here we describe an optical technique that sensitively and selectively detects flavor textures via the exciton response of a proximal transition metal dichalcogenide layer. Through a systematic study of rhombohedral and rotationally faulted graphene bilayers and trilayers, we show that when the semiconducting dichalcogenide is in direct contact with the graphene, the exciton response is most sensitive to the large momentum rearrangement of the Fermi surface, providing information that is distinct from and complementary to electrical compressibility measurements. The wide-field imaging capability of optical probes allows us to obtain spatial maps of flavor order with high throughput, and with broad temperature and device compatibility. Our work helps pave the way for optical probing and imaging of flavor orders in flat band graphene systems.

Flatband graphene systems provide a versatile platform for engineering correlated and topological phenomena. While their phase diagrams vary remarkably with sample configuration parameters such as layer number and relative alignment[1-21], a feature common to all systems is that small changes in the carrier density and other experimental tuning parameters drive flavor order transitions (FTs) in which the relative occupation of the (nominally degenerate) electron orbitals with differing spin and valley polarization changes. In both crystalline and twisted graphene systems, these transitions are often accompanied by superconducting domes, suggesting that flavor symmetry breaking may play an important role in superconducting pairing[9-11,22-24]. To refine the understanding of the phase diagram, FTs have been investigated by various experimental techniques including electrical transport[7-9,15-17], measurements of the thermodynamic compressibility and magnetization[7,9,17,25,26], and scanning tunneling microscopy[15,27-29]. However, these techniques all come with drawbacks: bulk electrical measurements typically fail in the face of spatial inhomogeneity and become less expressive in superconducting states, while scanning tunneling measurements are incompatible with the common dual-gated geometry. Moreover, these measurements are inherently low bandwidth, precluding studies of dynamics.

Here we describe an optical technique that addresses some of these challenges. Figure 1a illustrates the device scheme, in which a WSe₂ sensor layer is placed in direct contact with a target graphene system. The short-range interaction between graphene and WSe₂ leads to a shift in the quasi-particle bandgap of the WSe₂ that depends on the flavor polarization of the graphene layer; this can be read out optically via reflection contrast (RC) spectra. As we detail below, our

¹Department of Physics, University of California at Santa Barbara, Santa Barbara, CA, USA. ²Department of Physics, University of Texas at Austin, Austin, TX, USA. ³Materials Science and Engineering Program, School of Engineering for Matter, Transport, and Energy, Arizona State University, Tempe, Arizona, USA. ⁴Research Center for Electronic and Optical Materials, National Institute for Materials Science, AZ, Tsukuba, Japan. ⁵Research Center for Materials Nanoarchitectonics, National Institute for Materials Science, AZ, Tsukuba, Japan. ✉e-mail: macd@physics.utexas.edu; jinchenhao@ucsb.edu

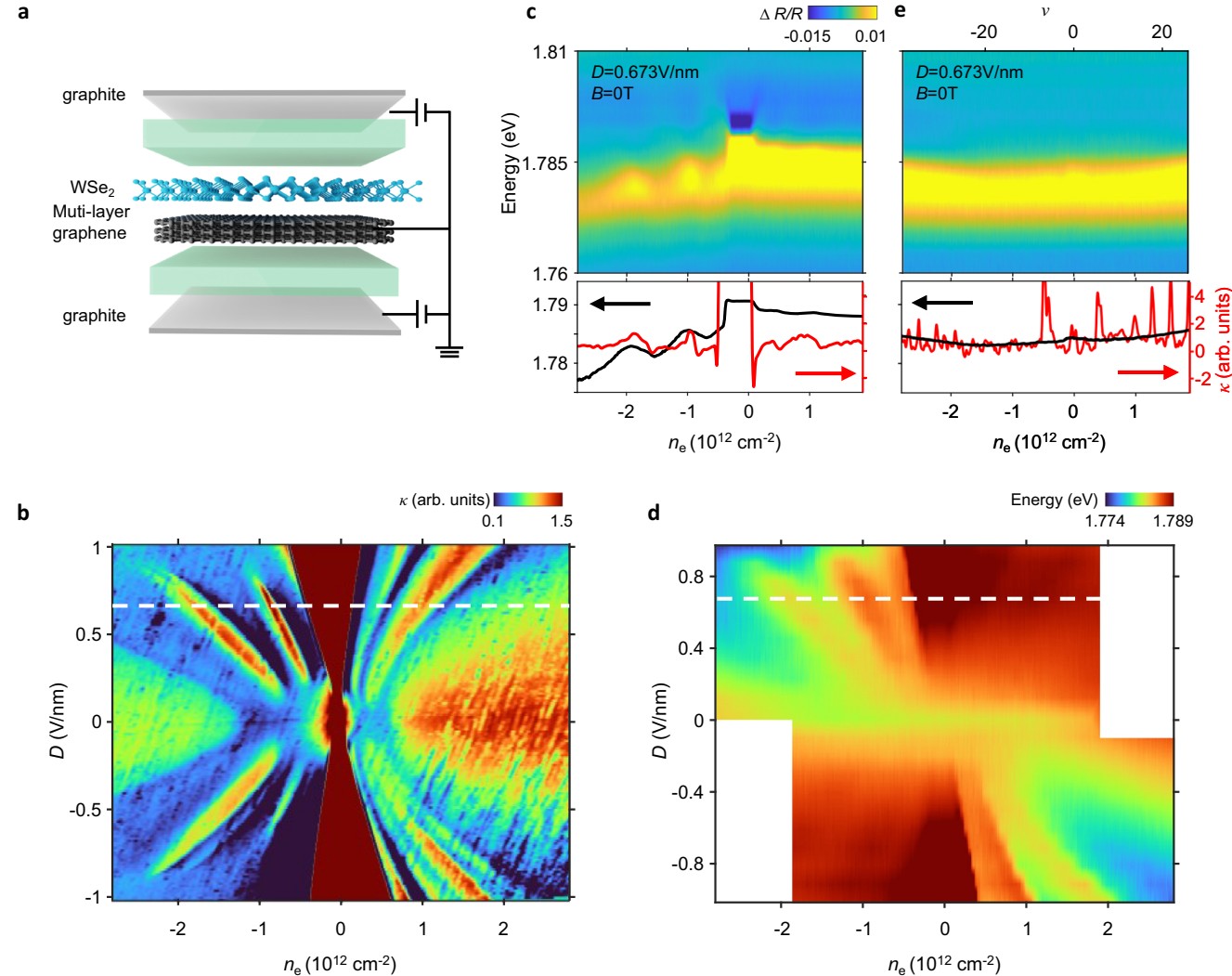

**Fig. 1 | Optical sensing of FT in RTG. a** Schematics of device configuration. A WSe$_2$ sensing layer is placed adjacent to flatband graphene without a spacer. The short-range interaction between graphene and WSe$_2$ imprints flavor orders of graphene into WSe$_2$ exciton responses. **b** Displacement-field and carrier-density dependent inverse compressibility of RTG device D1. **c** Upper panel: RC of device D1 at $D = 0.673$ V/nm and $B = 0$ T (white dotted line in (**b**, **d**) near WSe$_2$ 2s exciton resonance. Lower panel: extracted 2s exciton energy (black) and comparison to inverse compressibility (red). The exciton energy shift fully captures FT on the hole side.

**d** Displacement-field and carrier-density dependent 2s exciton energy of device D1. Features are only observed in the top-left and bottom-right quadrants owing to the sensitivity to layer polarization. **e** Upper panel: RC of device D1 at $D = 0$ V/nm and $B = 3$ T. Lower panel: extracted 2s exciton energy (black) and comparison to inverse compressibility (red). The prominent inverse compressibility peaks from charge gaps do not show up in optical sensing, in contrast to the FT in (**c**). All measurements are performed at a temperature of 3 K.

measurement configuration provides information that is distinct from electrical compressibility measurements or exciton sensing using a physically separated WSe$_2$ layer, which detects the small-wavevector limit of the polarzibility[30,31], and so offers us the capability in identifying flavor transitions.

## Results

### Optical sensing of FT

We first study a rhombohedral trilayer graphene (RTG) device D1. Figure 1b shows the inverse compressibility of the device measured at 3 K (see "methods"). The phase diagram features spontaneous formation of flavor orders on both electron and hole doping sides, consistent with previous reports[7,10]. Owing to sample inhomogeneity, two sets of patterns can be observed that are offset from each other (see Supplementary Fig. 1). Figure 1c shows RC of the same sample measured at displacement field $D = 0.673$ V/nm (dashed line in Fig. 1b). We focus on the spectral range near the 2s exciton resonance of WSe$_2$ (see Supplementary Fig. 2 for full spectra). The 2s exciton energy shows two

prominent kinks on the hole doping side, reminiscent of the two low compressibility lobes in Fig. 1b. The lower panel in Fig. 1c compares the fitted 2s exciton energy and the inverse compressibility (see "methods"). The optical spectrum reproduces all features of compressibility on the hole-doping side, while the response on the electron side is rather weak. This asymmetry is natural. Under the large displacement field, doped holes and electrons primarily reside in the top and bottom graphene layers, respectively. The much weaker WSe$_2$ response to electrons than holes indicates that WSe$_2$ primarily interacts with charges in the top (closest) layer with an interaction range $\Delta r < 1$ nm. Our sensing scheme therefore also provides a sensitive probe of layer polarization. Figure 1d summarizes the 2s exciton energy over similar parameter range as Fig. 1b (see "methods"). The optical phase diagram matches well with the electrical one, except that features appear only in the top left and bottom right quadrants owing to the layer polarization sensitivity.

Having demonstrated the ability to detect FT, we now show that our technique provides distinct information. Figure 1e compares

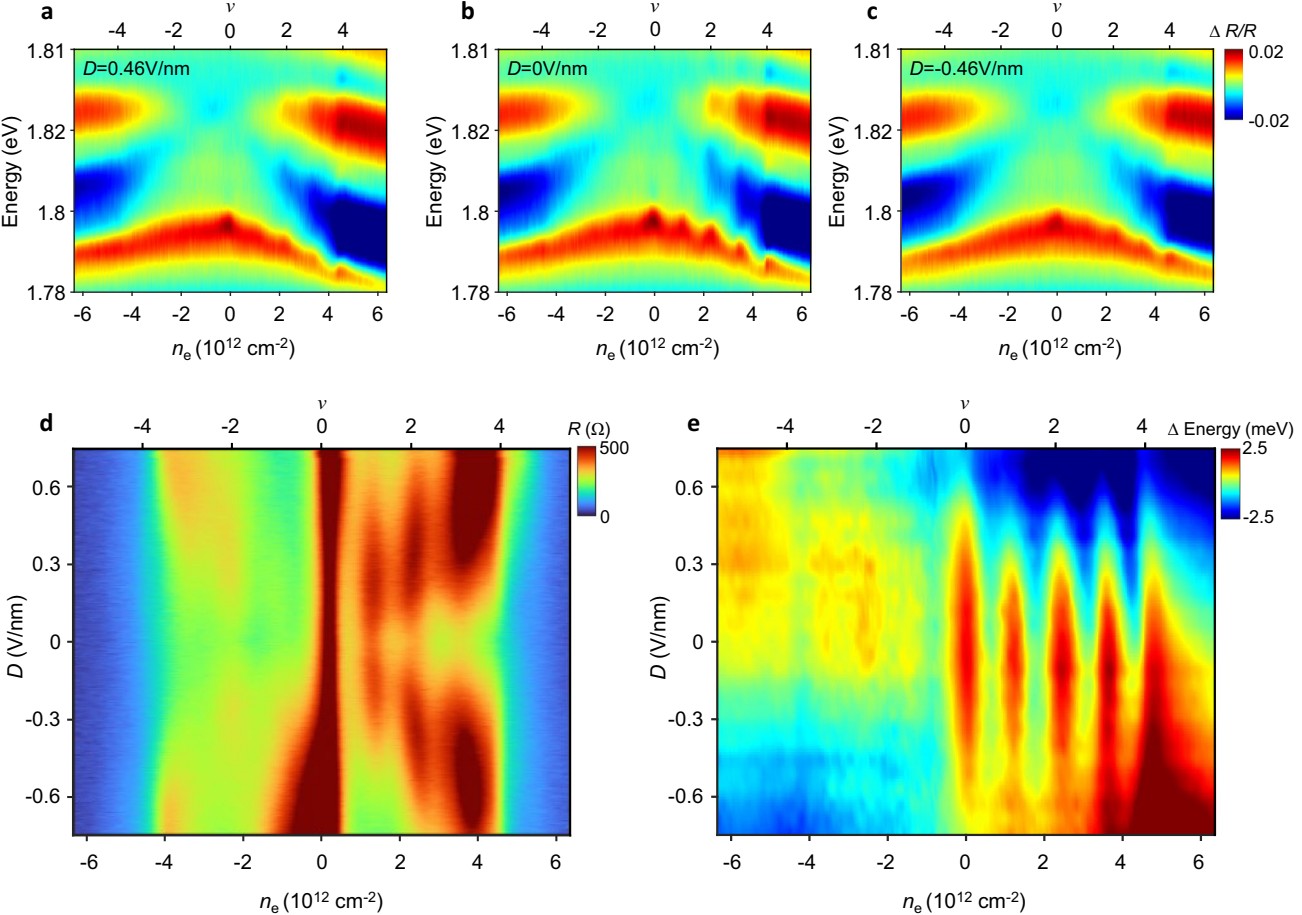

**Fig. 2 | Selective sensing of FT. a–c** RC of MATTG device D2 at $D$ = **a** 0.46, **b** 0, and **c** −0.46 V/nm. The 2s exciton resonance shows cascade features at integer fillings, which becomes weaker at larger displacement field. **d, e** Longitudinal resistance (**d**) and 2s exciton energy (**e**) of device D2 as a function of displacement field and carrier density. The insulating features at integer fillings in transport measurements become more prominent at larger displacement field due to the transition from Fermi surface resets to charge gaps. In contrast, the optical sensing is more sensitive to FT-induced Fermi surface reconstruction at low displacement field than the charge gaps at high displacement field. All measurements are performed at a temperature of 3 K.

electrical capacitance and optical RC measurements under the same experimental condition of $D$ = 0 V/nm and $B_z$ = 3 T (see Supplementary Fig. 1 for more results). A series of incompressible peaks emerge in capacitance, corresponding to gaps between Landau levels. Surprisingly, none of them appear in RC spectra. In capacitance the incompressible peaks are quite prominent, several times larger than the FT-induced features (Fig. 1c). If the optical response were effectively measuring compressibility one would expect, in contradiction to our observations, similarly strong features.

To gain further insight, we apply our sensing technique to alternating-twist magic-angle trilayer graphene (MATTG). Figure 2d shows the four-probe longitudinal resistance of MATTG device D2 with twist angle of 1.43° (see "methods"). The phase diagram is qualitatively consistent with previous reports[11,14,15,32] in that resistive states emerge at integer moiré fillings from $\nu$ = 0 to 4 under large displacement fields ($\nu$ = 1 corresponds to one electron per moiré period). At smaller displacement fields, the resistive behaviors at integer fillings become weaker, suggesting Fermi surface resets instead of gaps[11,15]. Interestingly, optical measurement of the same device shows the opposite trend. At zero displacement field (Fig. 2b), the 2s exciton resonance shows prominent cascade features at integer fillings, which becomes weaker at larger displacement fields (Fig. 2a, c). See Supplementary Fig. 3 for more data. Figure 2e summarizes the 2s exciton energy across the entire phase diagram. While

the emergence of features around integer fillings is consistent with transport measurement, their displacement field dependencies are in sharp contrast. Optical sensing does not weigh the gaps at large displacement field heavily but is sensitive to FT-induced Fermi surface reconstructions.

We have also performed measurements on Bernal bilayer graphene (BBG) in the quantum Hall regime. The upper panel in Fig. 3a shows the RC spectra of BBG device D3 under $B_z$ = 3 T and zero displacement field. The lower panel shows a comparison between the 2s exciton energy (black) and inverse compressibility (red) under the same measurement conditions. See Supplementary Figs. 4 and 5 for more results. A series of chemical potential jumps, observed as peaks in inverse compressibility, appear at even filling factors $\nu \in (-4, 4)$ and at higher filling factors only at the cyclotron gap filling factors, which are multiples of 4 because of spin-valley degeneracy. The peaks within $\nu \in (-4, 4)$ are related to flavor ferromagnetism. The optical measurement again shows quite distinct behavior. Instead of having features at even filling factors, the 2s exciton energy oscillates rapidly in the $\nu \in (-4, 4)$ interval between minima at odd filling factors and maxima at even filling factors. No strong features are seen at higher filling factors, even when the Fermi level lies in a cyclotron gap. The lack of gap features at high filling factors is consistent with our observations in RTG (Fig. 1e) and MATTG (Fig. 2e), and this makes the prominent features when $\nu \in (-4, 4)$ even more surprising.

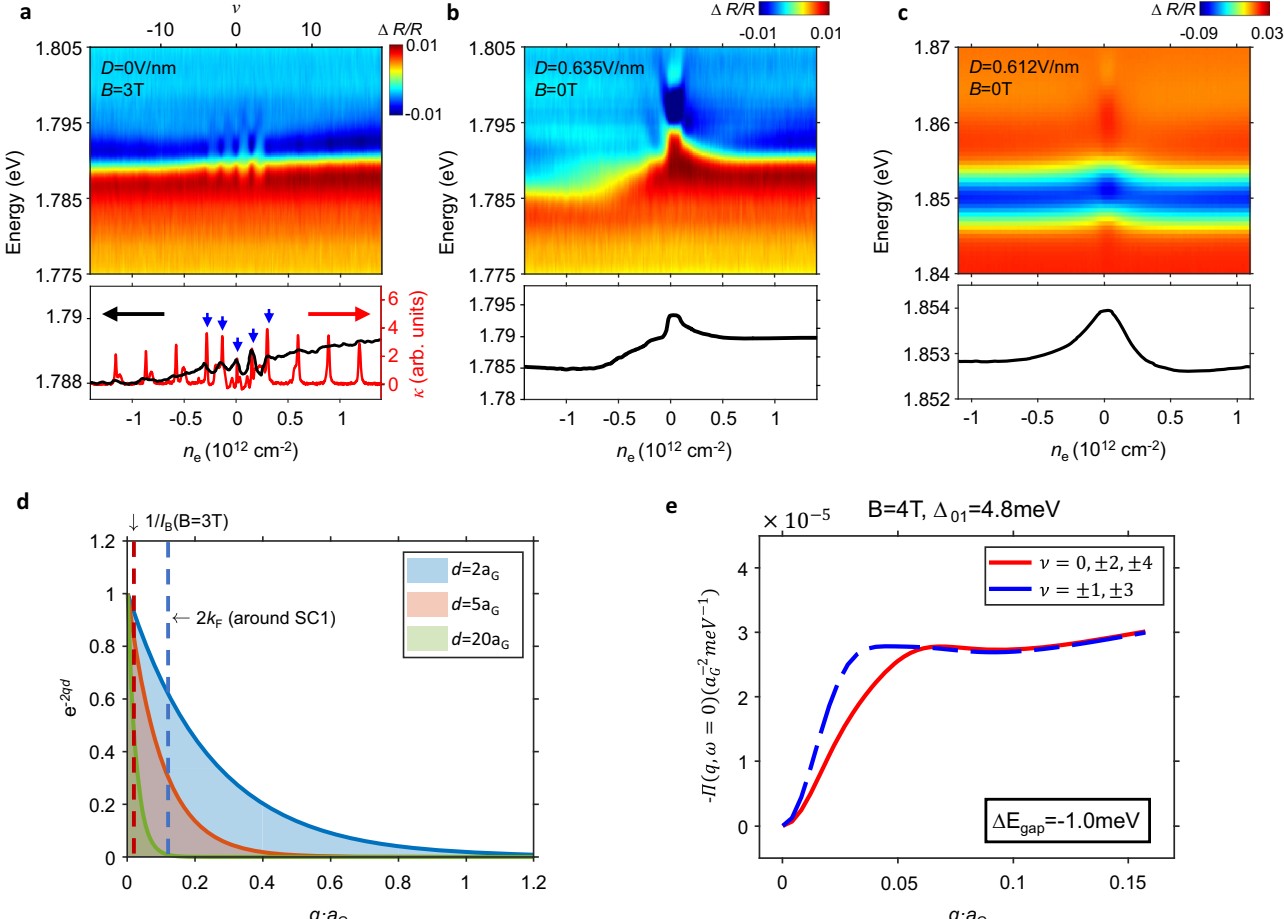

**Fig. 3 | Probing orbital polarization in BBG. a** Upper panel: RC of BBG device D3 at $D = 0$ and $B_z = 3$ T. Lower panel: extracted 2s exciton energy (black) and comparison with inverse compressibility (red). The strong inverse compressibility peaks from cyclotron gaps do not show up in RC. Instead, an oscillation of 2s exciton energy is observed between even and odd fillings within the zeroth Landau level. Blue arrows mark even fillings within the octet of the zeroth Landau level. **b, c** Comparison between device D3 without an hBN spacer (**b**) and BBG device D4 with a - 5 nm hBN spacer (**c**) under similar measurement configurations. Their distinct behaviors indicate the dominance of short-range and long-range interactions, respectively, as detailed in the text. **d** Momentum-cutoff for three representative interlayer distances $d$. $a_G = 0.246$ nm is the graphene lattice constant. Vertical dashed lines mark the momentum range of polarizability change from a cyclotron gap at $B_z = 3$ T (inverse magnetic length $q_B$, red) and from a representative FT in RTG (Fermi momentum $k_F$, blue). **e** Calculated static polarizability $\Pi(q, \omega = 0)$ of graphene at even (red) and odd (blue) Landau level fillings under magnetic field $B = 4$ T. The finite-$q$ part of graphene polarizability is enhanced at odd filling factors when $n = 0$ and $n = 1$ orbitals are alternately occupied, which can be uniquely accessed in adjacent layer exciton sensing as an energy shift $\Delta E_{gap}$. $\Pi(q, \omega) \approx \Pi(q, 0)$ for $\omega \ll \Delta_{01} = 4.8$ meV, where $\Delta_{01}$ is the orbital splitting.

## Selective detection of FT

Our investigations across multiple flat band graphene systems indicate that the optical sensing technique here provides qualitatively different information from electrical measurements and has unique FT sensitivity. Our results also contrast with those from the common exciton-sensing configuration with a thick hBN spacer, which largely reproduces electrical compressibility measurements, e.g. in the detection of graphene Landau levels[31]. To elucidate the origin of FT sensitivity, it is helpful to examine the role of an hBN spacer. To this end, we compare the RC spectra of device D3 and another BBG device D4 with a WSe$_2$ sensor layer and a 5 nm hBN spacer, as shown in Fig. 3b, c. Under similar measurement conditions, we observe two major differences. First, the 2s exciton without a hBN spacer appears at a much lower energy (Fig. 3b), indicating much stronger interaction between WSe$_2$ and graphene. Second, at large displacement field, the exciton responses without a hBN spacer show prominent asymmetry (Fig. 3b) between electron and hole doping while those with a spacer remain largely symmetric (Fig. 3c). The lack of layer sensitivity in the latter case suggests that the interaction is long-range in nature, which also explains the much weaker interaction strength. We therefore conclude

that the WSe$_2$-graphene interactions without (with) an hBN spacer are dominated by strong (weak) short- (long-range) interactions.

We have formulated a quantitative theoretical interpretation of our adjacent layer exciton sensing, one that also sheds light on the distinct information supplied by exciton sensing with hBN spacers. Our starting point is the successful GW theory of excitons[33,34] in TMDs (the "sensing layer"), within which the influence of a nearby 2D material (the "target layer") with negligible hybridization is captured exactly by adding a screening correction to Coulomb interactions $V_C \rightarrow V_C + \chi V_D^2$. Here $\chi$ is the target layer density-density response function, $V_D = 2\pi e^2 e^{-qd}/q$ is the interlayer Coulomb interaction and $d$ is the layer separation. $\chi V_D^2$ captures the contribution to the interaction between two electrons in the sensing layer that is mediated by charge density response in the target layer. Due mainly to reduced dimensionality, the 2s exciton has a rather small binding energy that is insensitive in absolute terms to screening[35-38] (see Supplementary Note 1 for a detailed discussion). Its resonance energy is then mainly determined by the quasi-particle bandgap of WSe$_2$. Because the carrier-density dependent part of the target layer response is at long wavelengths compared to the graphene lattice constant, the

quasiparticle bandgap change reduces to a simple exchange correction (see Supplementary Note 6). When the GW approximation is used for $\chi$, the quasiparticle band gap $E_{gap}$ is given by

$$E_{\mathrm{gap}} \rightarrow E_0 + \int \frac{d^2 q}{(2\pi)^2} \frac{\Pi_{22}\left(\boldsymbol{q}; \omega = \frac{\hbar^2 q^2}{2m^*}\right) V_D(q)^2}{1 - V_S(\boldsymbol{q})\Pi_{22}\left(\boldsymbol{q}; \omega = \frac{\hbar^2 q^2}{2m^*}\right)} \quad (1)$$

where $E_0$ is the quasi-particle bandgap of bare $WSe_2$, $\Pi_{22}$ is the single-particle polarization function, $m^*$ is the $WSe_2$ valence band effective mass and $V_S = 2\pi e^2/q$. An hBN spacer increases $d$ and decreases the momentum cutoff in the integral to $q_c < 1/d$. Figure 3d illustrates such momentum cutoff implied by $V_D$ for three representative interlayer distance that corresponds to the adjacent graphene layer ($d = 2a_G$), the distant graphene layer in RTG ($d = 5a_G$), and the case with a 5 nm hBN spacer ($d = 20a_G$), respectively. The small range of relevant momentum in the spacer case (green) captures the property that large-$q$ charge fluctuations do not produce a significant electrical potential in a distant layer. In the limit of thick hBN spacer and large $d$, $q_c \rightarrow 0$. $E_{gap}$ then depends on the graphene polarizability in the long wavelength and static limit, which is directly related to its compressibility. Therefore, the exciton sensing scheme with an hBN spacer largely reproduces results from electrical measurements.

The case without an hBN spacer is distinctively different since $d \sim 0.5\,nm$ for the closest graphene layer. $E_{gap}$ senses changes in the large-$q$ parts of graphene polarizability, which typically dominate due to their larger phase space (Fig. 3d). Our sensing scheme therefore mainly probes the large-$q$ polarizability of graphene, which is inaccessible to electrical measurements. The observed layer sensitivity (Fig. 1) is a direct manifestation, where the bandgap shifts induced by the adjacent and distant graphene layers in RTG differ by several times. As illustrated in Fig. 3e, the difference between the two cases (blue and red) only becomes prominent at large-$q$. The much larger bandgap shift from the adjacent layer confirms the dominance of large-$q$ contribution. This unique capability allows it to identify physics unrelated to the appearance of charge gaps, such as a cyclotron gap, that mainly affects the small-$q$ part of polarizability (red dashed line). It also explains its sensitivity to FT since FT involves reconstruction of the entire Fermi surface and modifies the large-$q$ polarizability up to several times of $k_F$ (blue dashed line). In Supplementary Note 6 we show that the 2s exciton energy changes that accompany FT in RTG (Fig. 1) agree quantitatively with the calculations.

The surprising exciton energy oscillations we have discovered in the small filling factor $\nu \in (-4, 4)$ regime of BBG provide another excellent example of our capability on sensing large-$q$ polarizability of graphene. As illustrated in Fig. 3e, we interpret the minima in the 2s exciton energy at odd filling factors as evidence for orbital-polarized states with differential occupation between the $n = 0$ and $n = 1$ orbitals, which lead to strong screening over a wide range of wavevectors from inter-orbital contribution (see Supplementary Note 6); and the maxima at even filling factors as evidence for states in which both orbitals of a given flavor are completely occupied or empty. The differences between even and odd fillings only appear at nonzero $q$ (Fig. 3d), therefore the oscillation shows up in optical sensing but not compressibility (Fig. 3a). The convenient optical probe of the orbital content of fractional states in the $\nu \in (-4, 4)$ regime of BBG could aid efforts to optimize robust non-Abelian quantum Hall states in the bilayer graphene platform[39–41].

### Wide field imaging of FT

Besides FT sensitivity, our technique also offers wide-field imaging capability to capture spatial patterns of FT with high throughput. Figure 4a, b shows the optical microscope image and reflection contrast spectra of a magic angle twisted bilayer graphene (MATBG) device D5. FT has been widely reported in MATBG, giving rise to Dirac

revivals and Chern insulators at integer fillings[1,26,42–47]. Indeed, we observe clear features in 2s exciton resonance at integer moiré fillings $\nu = \pm 1$ to $\pm 4$ (orange arrows)[48,49]. On the other hand, MATBG is known for its high sensitivity to twist angle and intrinsic spatial inhomogeneity from lattice relaxation. Figure 4c shows reflection contrast on a different spot in the same device, where we only observe the band insulator at $\nu = \pm 4$ but no features in between. In transport measurement of this device (Supplementary Fig. 6), we consistently observe strongly insulating states at $\nu = \pm 4$, while the features at $\pm 1$ to $\pm 3$ are generally weak and inconsistent between different source-drain configurations. These observations exemplify a common challenge plaguing the study of twisted graphene systems, where devices vary strongly and it can often be difficult to extract intrinsic physics[3,26]. For example, different transport phenomena can be dominated by different conducting channels and may not be directly correlated.

Our technique offers a potential solution. As a demonstration, we perform wide-fielding imaging of the $\nu = 4$ band insulator and $\nu = 2$ cascade feature in Fig. 4d, e, respectively. This allows us to extract a spatial map of twist angle from the charge density at $\nu = 4$, and a map of the correlation strength from the prominence of cascade feature at $\nu = 2$. Each map is obtained in 15 min without spatial scanning (see "methods" and Supplementary Fig. 7). The cascade features only appear in a small spatial region close to the left edge of this device, which explains the weak and inconsistent features in transport. By comparing the FT map and angle map, we find that the cascade features emerge in a twist angle range between 1.01 and 1.07 ° and are the most prominent at angles around 1.04°. See Supplementary Note 5 for more discussions on potential strain effects.

The high-throughput imaging capability of our technique is further augmented by its broad environment compatibility. It encodes low-energy flavor physics into exciton responses at a much higher energy scale and is less susceptible to noise. Figure 4f shows the 2s exciton energy at different temperatures up to 50 K. Exciton resonances remain largely unchanged over this temperature range, allowing us to directly track melting of the cascade features. Interestingly, the cascade features remain visible at 50 K, consistent with previous reports from chemical potential measurements[25,26,50,51] and is an order of order of magnitude higher than the temperature at which hysteresis of isospin ferromagnetism disappears[1,4,42,43]. This may suggest the existence of vestigial FT or flavor fluctuations over a broad temperature range[25,50] (see Supplementary Note 4 for more discussions).

## Discussion

Moiré graphene multilayers have been previously reported to remotely "imprint" a superlattice potential in an adjacent $WSe_2$ layer and generate exciton replicas[48,49,52]. On the other hand, optical studies of flavor physics in flatband graphene remain largely unexplored, especially in non-moiré systems. The technique reported here applies to both moiré and crystalline graphene and opens several exciting opportunities in studying flavor orders, transitions, and their interplay with other correlated phases (see Supplementary Note 3 for more discussions). It offers an attractive approach to disentangle large-$q$ changes of graphene polarizability, such as flavor orders and fluctuations, from local Fermi surface distortion, such as single particle gaps, nematicity and charge density waves[53–57], thereby shedding light on the roles of these instabilities. The high-throughput imaging capability, along with the wide temperature and device geometry compatibility, enables investigation of FT spatial patterns near and across critical points. A particularly exciting opportunity lies in in-situ imaging of FT throughout the superconductivity domes in magic angle multi-layer graphene[22,23], which can be correlated to transport measurements to disentangle extrinsic and intrinsic effects and potentially elucidate the interplay between FT and superconductivity. By establishing an optical

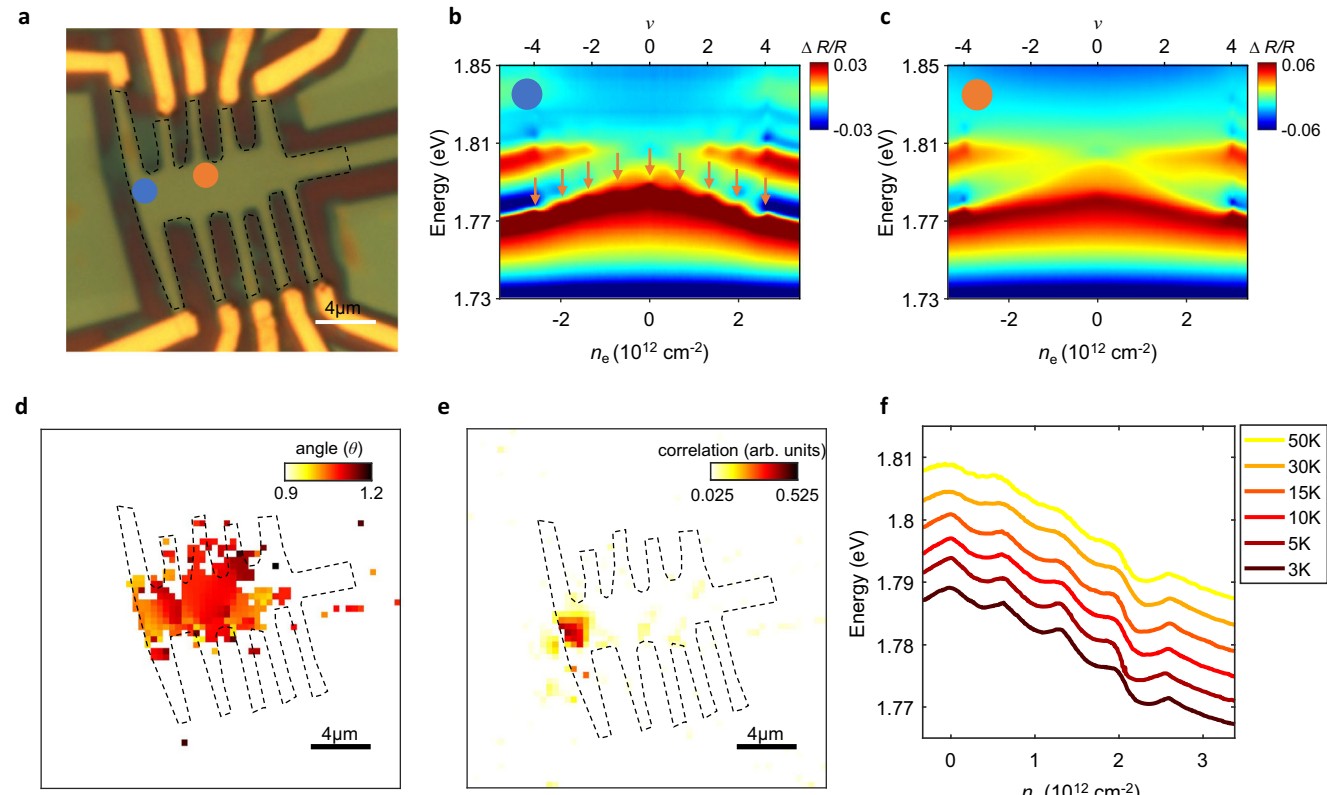

**Fig. 4 | Wide field imaging of FT. a** Optical microscope image for MATBG device D5. Scale bar: 4μm. **b,c** RC of representative magic-angle (**b**) and non-magic angle (**c**) spots in device D5 with local twist angle of 1.04° and 1.14°, respectively. Their locations are marked by blue and orange dots in (**a**). **d**, **e** Spatial map of twist angle (**d**) and correlation strength (**e**) extracted from the $v = 4$ and $v = 2$ cascade features, respectively. Both maps are obtained by wide-field imaging without scanning. **f** Temperature dependence of the extracted 2s exciton energy. The cascade features persist to above 50 K.

technique to detect FT, our work also paves the way for dynamic manipulation and investigation of flatband graphene systems using ultrafast light pulses, such as Floquet engineering of FT and studying its non-equilibrium dynamics.

# Methods

## Sample fabrication

The preparation of multilayer graphene, hexagonal boron nitride (hBN), and tungsten diselenide (WSe₂) flakes involves mechanical exfoliation of bulk crystals onto silicon substrates with a 285 nm silicon oxide layer. Rhombohedral domains within trilayer graphene flakes are identified using a Horiba T64000 Raman spectrometer equipped with a 488-nm mixed-gas Ar/Kr ion laser beam. Subsequent isolation of the rhombohedral domains is performed utilizing a Dimension Icon 3100 atomic force microscope[58,59].

All Van der Waals heterostructures are constructed through a standard dry-transfer technique employing a poly (bisphenol A carbonate) (PC) film on a polydimethylsiloxane (PDMS) stamp. The fabrication process involves initially creating the lower hBN/graphite part, releasing them onto a 90 nm Si/SiO₂ substrate. The removal of polycarbonate residue on the sample is accomplished by dissolving it in chloroform, followed by rinsing with isopropyl alcohol and annealing at 375 °C. The upper part of the heterostructure is separately assembled and transferred onto the lower part. This stacking sequence is meticulously implemented to minimize mechanical stretching of the multilayer graphene. Standard electron-beam lithography, dry-etching processes, and vacuum deposition are employed to fabricate electrodes for electrical contacts (~150 nm gold with ~5 nm chromium and ~15 nm palladium adhesion layers).

## Calibration of carrier density, displacement field, and twist angle

Carrier densities in all devices are calibrated from the hBN thickness measured by a Dimension Icon 3100 atomic force microscope. Using hBN dielectric constant $\underline{\varepsilon}_{hBN} = 3.52$, we compute the geometrical capacitance per unit area $c_{t,b} = \varepsilon_{hBN}\varepsilon_0/d_{t,b}$ between the top/bottom gate and sample, where $d_t$ ($d_b$) is the top (bottom) hBN thickness. The charge density and displacement field are obtained as $n_0 = (c_tV_t + c_bV_b)/2e$ and $D = (c_tV_t - c_bV_b)/2\varepsilon_0$, respectively, where $V_t$ ($V_b$) is the top (bottom) gate voltage and $e$ is elementary charge.

The twist angle of MATBG and MATTG are extracted from the cascade features at superlattice filling factors $v = \pm 4$ (Fig. 2 and Fig. 4). From the corresponding carrier density $n_{v=4}$, the twist angle $\theta$ was obtained from $n_{v=4} = (8\theta2)/(\sqrt{3}a_02)$, $a_0 = 0.246$ nm is the graphene lattice constant.

## Reflection contrast (RC) measurement

The devices were mounted in a closed-cycle cryostat (Quantum Design, OptiCool) for all optical experiments with a base temperature of 3 K. A broadband tungsten lamp was beam-shaped by a single mode fiber and subsequently collimated by a lens. The light was focused onto the sample by an objective (NA = 0.45), resulting in a beam diameter of ~1 μm on sample with a power of ~20 nW. The reflected light was collected by a liquid-nitrogen-cooled CCD camera coupled with a spectrometer. The reflection contrast was computed as RC = (R′ − R)/R, where R′ and R represent the reflected light intensity from regions with and without the sample, respectively. Keithley 2400 source meters were employed to apply gate voltages to adjust the charge density.

### Extraction of 2s exciton energy

We extracted the 2s exciton energy at each carrier density from the local maximum in the slope of RC vs. probe energy (Supplementary Fig. 2a and 2c). The obtained 2s exciton energy shows a smoothly decreasing background with increasing charge density due to stronger screening. This background dominates the exciton energy shift in MATTG owing to the large range of carrier density. To highlight the cascade features associated with the FT, we fitted the smooth background for the hole (electron) side using a 3rd (7th)-order polynomial (Supplementary Fig. 2d, Orange curve). The background-subtracted 2s exciton energy shows clear cascade features at integer fillings (Supplementary Fig. 2d). The same background was used for all displacement fields to ensure that no artifacts were introduced (Fig. 2e).

### Capacitance and transport measurement

Penetration field capacitance measurements were performed on WSe$_2$/RTG device D1 and WSe$_2$/BBG graphene devices D3. The device capacitance $c_p$ was isolated from the environment using a low-temperature capacitance bridge[60]. The inverse compressibility $\kappa$ was obtained from $c_p$ through $c_p = c_t c_b/(c_t + c_b + \kappa^{-1}) \approx \kappa c_t c_b$[61]. The magnitude of $\kappa$ increases when the sample is incompressible (gapped) and decreases when it is compressible (conducting). The measurement of $\kappa$ involved applying a fixed AC excitation (17–88 kHz) to the top gate. The phase and amplitude of a second AC excitation of the same frequency were adjusted and applied to a standard reference capacitor ($c_{ref}$) on the low-temperature amplifier to balance the capacitance bridge. A commercial high-electron-mobility transistor (FHX35X) transformed the small sample impedance to a 1 kΩ output impedance, yielding a gain of about 1000. The DC components of $V_t$ and $V_b$ were supplied by Keithley 2400 source meters and were connected to the corresponding gate though bias tee. Additional electrodes were patterned in WSe$_2$/MATTG device D2 and WSe$_2$/MATBG device D5 for electrical transport. Four-point longitudinal resistance was obtained by supplying an AC current of 10 nA amplitude at frequency of 17.777 Hz.

### Widefield imaging of cascade features

A broadband supercontinuum laser (YSL photonics SC-OEM) was filtered by a home-built double monochromator to generate probe light of tunable center wavelength and <0.2 nm full width at half maximum (FWHM). The probe light was expanded before focusing on the sample, giving rise to a field of view of ~ 900 μm$^2$ that covers the entire device. The wide-field image of sample was collected by an EMCCD camera (ProEM-HS 512BX3) without spatial scanning. To obtain a map of the cascade features, we tuned the probe light energy to be slightly above the WSe$_2$ 2s exciton resonance and took a sample reflection image at each carrier density. Ordinarily, the 2s exciton energy redshifts with increasing carrier density, leading to decrease of sample reflection at the probe energy. On the other hand, the cascade features at integer fillings lead to abnormal blueshifts of 2s exciton energy with increasing carrier density (Fig. 4b) and thereby increase of sample reflection. This allowed us to extract both carrier density and strength of the cascade features by comparing sample images at neighboring carrier density.

We further employed a lock-in algorithm to improve the signal to noise ratio. The carrier density in the device was modulated at 66 Hz by a small AC gate voltage $\Delta V_g = 0.01$ V on top of the DC gate voltage $V_g$. The EMCCD camera was externally triggered and synchronized with the AC gate modulation, thereby directly obtaining the differential reflection image of the sample between slightly different carrier densities. Supplementary Movies 1 and 2 show the obtained differential reflection images for a range of carrier densities near $\nu = 2$ and $\nu = 4$ of MATBG, respectively. Supplementary Fig. 7 shows the carrier density-dependent differential reflection near $\nu = 4$ for a representative spatial

spot (blue boxed pixel). The non-monotonic dip from the cascade features was fitted by a 2nd-order polynomial, from which we extracted the carrier density and the amplitude of the $\nu = 4$ cascade feature. Similar fitting was performed on each pixel for carrier densities near $\nu = 4$ and $\nu = 2$, from which we obtained a map of the twist angle and correlation strength (Fig. 4d, e).

## Data availability

Data in the main text and Supplementary fig. 1–8 are available on Open Science Framework[62]. All other data that supports the findings of this study are available from the corresponding authors upon request.

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

## Acknowledgments

We acknowledge the use of the research facilities within the California NanoSystems Institute, supported by the University of California, Santa Barbara and the University of California, Office of the President. C.J. acknowledges support from Air Force Office of Scientific Research under award FA9550-23-1-0117. The sample fabrication is supported by the National Science Foundation through a CAREER award DMR-2337606. T.X. acknowledges the support from the National Science Foundation Graduate Research Fellowship under Grant No.2139319. A.F.Y. acknowledges primary support by the Department of Energy under award DE-SC0020043. A.F.Y. acknowledges the support of the Gordon and Betty Moore Foundation under award GBMF9471 and the Packard Foundation under award 2016-65145 for general group activities. This work made use of facilities funded by Enabling Quantum Leap: Convergent Accelerated Discovery Foundries for Quantum Materials Science, Engineering and Information (Q-AMASE-i) award number DMR-1906325 from the National Science Foundation. A.H.M. and T.M.W. acknowledge financial support from the NSF (Award No. DMR–2308817 and DMR–2308817). S.T. acknowledges primary support from DOE-SC0020653 (materials synthesis), DMR 2111812, DMR 2206987 and CMMI 2129412 (manufacturing). S.T. acknowledges support from Lawrence Semiconductor Labs. K.W. and T.T. acknowledge support from the JSPS KAKENHI (Grant Numbers 20H00354 and 23H02052) and World Premier International Research Center Initiative (WPI), MEXT, Japan.

## Author contributions

C.J. conceived and supervised the project. S.X., Z.C. and T.X. fabricated the device. T.X. and R.X. performed the optical measurements. L.F.H., Y.G., O.I.S., C.P. and T.A. performed the electrical measurements. T.X. analyzed the data. T.M.W. and A.H.M. contributed to the theoretical interpretation and performed numerical simulations. Y.O., P.H. and S.A.T. grew the WSe₂ crystals. K.W. and T.T. grew hBN crystals. C.J., A.H.M. and A.F.Y. wrote the paper with input from all the authors.

## Competing interests

The authors declare no competing interests.
