## [Transparent Peer Review file · Nature Communications]

Optical Imaging of Flavor Order in Flat Band Graphene

Corresponding Author: Professor Chenhao Jin

Version 0:

Reviewer comments:

Reviewer #1

(Remarks to the Author)

The authors have effectively addressed my comments as well as those from other referees, and I am satisfied with nearly all of their responses. However, I cannot agree with their statement regarding the validity of the Rytova-Keldysh potential. Estimates indicate that the screening radius r_0 in the Rytova-Keldysh potential exceeds the layer thickness. If this is not the case, its applicability within the framework of microscopic electrodynamics is questionable. I recommend that the authors calculate the product $k_F r_0$ (where k_F is the Fermi wavevector). If this product is sufficiently large, they should reconsider their analysis.

Reviewer #2

(Remarks to the Author)

I'm happy with the changes and the rebuttal of the authors to my questions and I think the paper is now ready to go.

Reviewer #3

(Remarks to the Author)

The authors have conducted a thorough analysis and provided additional evidence supporting their argument on flavor order sensing via the large- q sensitivity of adjacent WSe_2 . This new interpretation of WSe_2 exciton sensing is novel and has the potential to advance applications of this technique. I appreciate the authors' efforts in improving the manuscript. One remaining question concerns the experimental temperature. If all measurements were performed at 3K, does this imply that certain flavor order persists in all these systems (BBG, MATBG, MATTG, RTG) at such a high temperature? Given that valley- or spin-polarized states are typically highly sensitive to temperature, I would expect them to be more fragile at 3K as in transport measurements. Does this suggest that the large- q component of these states exhibits a higher critical temperature? Furthermore, are there any valley or spin polarized features in WSe_2 optical response at this temperature, for example, in the helicity resolved reflectance?

Version 1:

Reviewer comments:

Reviewer #1

(Remarks to the Author)

I got the point of the authors regarding the use of the Rytova-Keldysh potential, thank you for clarifying. I still think that a self-consistent solution for a multilayer system with account for the spatial inhomogeneity would be instructive, but for a relatively short manuscript like this one the treatment seems to be OK. I recommend accepting the manuscript.

Reviewer #3

(Remarks to the Author)

The authors have addressed my concerns and I recommend the manuscript for publication in Nature Communications.

Reviewer #1 (Comments for the Author):

The authors have effectively addressed my comments as well as those from other referees, and I am satisfied with nearly all of their responses. However, I cannot agree with their statement regarding the validity of the Rytova-Keldysh potential. Estimates indicate that the screening radius r_0 in the Rytova-Keldysh potential exceeds the layer thickness. If this is not the case, its applicability within the framework of microscopic electrodynamics is questionable. I recommend that the authors calculate the product $k_F r_0$ (where k_F is the Fermi wavevector). If this product is sufficiently large, they should reconsider their analysis.

Reply R1: We thank the Referee for the suggestion to further clarify this issue. Indeed, as the reviewer pointed out, the typical TMD screening length $r_0 \sim 30$ to 80 \AA is much larger than the layer thickness $d_{\text{TMD}} \sim 6 \text{ \AA}$ ¹. This is because the TMD layer has a much larger dielectric constant than the surrounding hBN. An effective R-K potential captures some relevant details of the dielectric screening environment, both TMD and hBN, in an approximate way that can be useful in certain contexts, such in reproducing the non-hydrogenic sequence of exciton binding energies in monolayer TMD¹. However, we believe the R-K potential is not suitable in our analysis owing to two major differences:

First, the bandgap shift (Eq. 1 in the main text) does not depend on intralayer Coulomb interaction in the TMD layer. This can be intuitively understood since the bandgap shift originates from interactions between electrons in the TMD sensor layer and the system of interest (graphene). Therefore, V_S and V_D in Eq. 1 correspond to intralayer Coulomb interaction in graphene and interlayer Coulomb interaction, respectively. While R-K potential has been used to describe intralayer interaction in the TMD layer, it has not been used to describe intralayer interaction in graphene or interlayer interactions². In particular, the background dielectric constant of graphene is rather small³ and not significantly different from the hBN environment.

Second, the 1s exciton has a small Bohr radius $\sim 1 \text{ nm}$, which is comparable to the layer thickness of TMD ($\sim 6 \text{ \AA}$). In contrast, the relevant k_F here is one order of magnitude smaller than inverse thickness of graphene ($\sim 3 \text{ \AA}$). Therefore, at these wavevectors the graphene layer can safely be regarded as thin, and the dielectric environment can be regarded as consisting entirely of the encapsulant material.

We have included the above discussion in the revised manuscript and updated the supplementary materials accordingly.

Reviewer #3 (Comments for the Author):

The authors have conducted a thorough analysis and provided additional evidence supporting their argument on flavor order sensing via the large- q sensitivity of adjacent WSe_2 . This new interpretation of WSe_2 exciton sensing is novel and has the potential to advance applications of this technique. I appreciate the authors' efforts in improving the manuscript. One remaining question concerns the experimental temperature. If all measurements were performed at 3K, does this imply that certain flavor order persists in all these systems (BBG, MATBG, MATTG, RTG) at such a high temperature? Given that valley- or spin-polarized states are typically highly sensitive to temperature, I would expect them to be more fragile at 3K as in transport measurements. Does this suggest that the large- q component of these states exhibits a higher critical temperature?

Reply R2: We thank the reviewer for this important question. It is indeed interesting that the flavor order features persist to high temperature in our optical measurements. We believe the observed features indicate the existence of short-range orders or strong fluctuations, not necessarily long-range orders.

Experimental probes of isospin orders in graphene can be divided into two types. The first type requires a single domain, such as Hall and SQUID measurements of magnetization. Two domains with opposite magnetization would give opposite signal in SQUID that cancels each other. In contrast, the second type does not require a single domain, such as chemical potential and compressibility measurements⁴. Our optical sensing belongs to the second type: the signal originates from dielectric screening, which is insensitive to the sign of order parameter. For example, two domains with opposite spins would give the same shape of Fermi surface (despite from different spins) and therefore the same dielectric function. Consequently, the existence of multiple domains does not affect the second type of measurements as long as the domains size is much larger than $1/k_F$ and the domain lifetime much larger than $1/E_F$, such that the Fermi surface is well defined.

When the temperature exceeds the critical temperature, long-range flavor order vanishes. On the other hand, there can still be short-range orders or strong fluctuations in the order parameters. They can be intuitively understood as spontaneously formed domains that keep fluctuating over space and time. Their signals would disappear in the first type of measurement but remain finite in the second type until the characteristic length/timescale of the fluctuating orders drop below the threshold above. Since the large- q part of polarizability is less sensitive to a finite domain size, we expect our optical sensing technique to be able to capture shorter-range orders and weaker fluctuations compared to zero- q measurements such as compressibility. For example, the anomalous Hall effect in RTG's quarter-metal region only survives up to 2.3 K. In contrast, their signatures remain visible at 6K in compressibility measurement and persists up to 15K in our optical sensing technique⁴.

In summary, our optical sensing technique captures both long-range orders and short-range fluctuations. Combined with its sensitivity to flavor orders, we expect it to open up new opportunities for studying fluctuating orders at elevated temperatures. We have included the above discussion in the revised manuscript.

Furthermore, are there any valley or spin polarized features in WSe_2 optical response at this temperature, for example, in the helicity resolved reflectance?

Reply R3: We thank the reviewer for the question. As detailed in reply R2, our optical sensing based on dielectric screening is not sensitive to the sign of order parameter. Therefore, we do not expect different responses in helicity-resolved reflectance. Fig. R1 shows reflection contrast of WSe₂/MATBG measured with left- and right-circularly polarized light. Indeed, the two configurations show identical spectra throughout the phase diagram.

Figure R1. (a) Helicity-resolved reflection contrast for MATBG device D5. (b) Comparison between the two configurations at $n_e = 2.61 \times 10^{12} \text{ cm}^{-2}$.

References

1. Wang, G. *et al.* Colloquium: Excitons in atomically thin transition metal dichalcogenides. *Rev Mod Phys* **90**, 021001 (2018).
2. Hwang, E. H. & Das Sarma, S. Dielectric function, screening, and plasmons in two-dimensional graphene. *Phys Rev B* **75**, 205418 (2007).
3. Wehling, T. O. *et al.* Strength of Effective Coulomb Interactions in Graphene and Graphite. *Phys Rev Lett* **106**, 236805 (2011).
4. Holleis, L. *et al.* Fluctuating magnetism and Pomeranchuk effect in multilayer graphene. *Nature* (2025) doi:10.1038/s41586-025-08725-5.